# Increased muscle satellite cell content and preserved telomere length in response to combined exercise training in patients with FSHD

Oscar Horwath[1] , Diego Montiel-Rojas[1] , Elodie Ponsot[1] , Léonard Féasson[2,3] and Fawzi Kadi[1]

[1]*School of Health Sciences, Örebro University, Örebro, Sweden*
[2]*Inter-University Laboratory of Human Movement Sciences, University Lyon, UJM-Saint-Etienne, Saint-Etienne, France*
[3]*Myology Unit, Referent Center of Rare Neuromuscular Diseases, Euro-NmD, University Hospital of Saint-Etienne, Saint-Etienne, France*

Handling Editors: Paul Greenhaff & Matthew Brook

The peer review history is available in the Supporting Information section of this article (https://doi.org/10.1113/JP287033#support-information-section).

**Abstract figure legend** Sixteen facioscapulohumeral muscular dystrophy (FSHD) patients were randomized into either a control- or a training group. The training program spanned over 24 weeks (3 days week$^{-1}$) and consisted of continuous aerobic training, strength training and high-intensity intervals. Muscle biopsies were obtained for histological and immunofluorescence analyses of slow and fast muscle fibres, as well as analyses of muscle telomere length. The exercise training program induced muscle fibre growth and satellite cell pool expansion in fast fibres. Importantly, training in these patients did not exacerbate the dystrophic phenotype because immune cell infiltration, markers of regeneration and muscle telomere length remained unaltered. Collectively, exercise training leads to positive adaptations in FSHD muscle in the form of muscle fibre growth and satellite cell pool expansion without altering the remaining regenerative capacity. These novel findings reinforce the notion that physical activity is a beneficial strategy for attenuating muscle tissue deterioration in patients with FSHD. Created using BioRender.com.

**Abstract** Facioscapulohumeral muscular dystrophy (FSHD) is an inherited muscle disease characterized by weakness and muscle wasting. In the absence of available treatments, exercise training has emerged as a potential strategy to attenuate muscle tissue deterioration. However, little is known about the impact of chronic exercise on degenerative events and regenerative capacity in FSHD muscle. Muscle biopsies were obtained from 16 FSHD patients before and after a 24 week training program combining aerobic-, strength- and high-intensity exercise (Control; $n = 8$, Training; $n = 8$). Histochemical and immunohistochemical approaches were applied to assess histopathological signs, markers of regeneration, inflammatory infiltrates and satellite cell content. Muscle telomere length was measured as an indicator of the remaining regenerative capacity. The proportion of muscle fibres expressing developmental myosins and centralized myonuclei was not exacerbated after the intervention. Similarly, no alterations were observed in the number of inflammatory infiltrates (CD68$^+$ cells). Alongside muscle hypertrophy in slow ($P = 0.022$) and fast fibres ($P = 0.022$ and $P = 0.008$), satellite cell content increased specifically in fast fibres (+75 %, $P = 0.015$), indicating a functional satellite cell pool in FSHD muscle. Importantly, exercise training was not associated with a shortening of muscle telomere length, suggesting that muscle cell turnover was not accelerated despite an expansion of the satellite cell pool. Our findings suggest that combined exercise training elicits beneficial muscular adaptations without impairing important indicators of skeletal muscle regenerative capacity in patients with FSHD.

(Received 31 May 2024; accepted after revision 17 January 2025; first published online 31 January 2025)

**Corresponding author** F. Kadi: School of Health Sciences, Örebro University, 701 82, Örebro, Sweden. Email: fawzi.kadi@oru.se

**Key points**

- A 24 week combined exercise training program is a safe and well-tolerated strategy to attenuate skeletal muscle deterioration in facioscapulohumeral muscular dystrophy (FSHD) patients.
- Markers of histopathology, muscle fibre regeneration and inflammatory infiltrates were not exacerbated following exercise training in FSHD muscle.
- Here, we show novel data that exercise training in FSHD patients induced muscle fibre hypertrophy and triggered an expansion of the satellite cell pool specifically in fast fibres.
- Exercise training in these patients is not associated with a shortening of muscle telomere length thereby indicating a preserved capacity for muscle regeneration.

## Introduction

Facioscapulohumeral muscular dystrophy (FSHD) is an autosomal-dominant neuromuscular disease, where the slow and progressive decline in muscle function, together with excessive fatigue, reduced physical activity and mobility impairment contributes to the deterioration of general health (Sacconi et al., 2015; Wang & Tawil, 2016). At the muscle level, fibre necrosis, fibrosis and immune cell infiltration have been documented, indicating the occurrence of ongoing muscle degeneration (Arahata et al., 1995; Banerji et al., 2020; Bodensteiner & Schochet, 1986). Although no pharmacological treatment is available to treat or delay the progression of FSHD,

**Oscar Horwath** received his bachelor's degree in exercise biomedicine from Halmstad University and his master's degree from Örebro University in physiology and medicine. Oscar currently works as a PhD student in the Åstrand Laboratory at the Swedish School of Sport and Health Sciences. His research is focused on the physiology of human skeletal muscle, with a special interest in the underlying mechanisms of muscle loss in the context of ageing and disease.

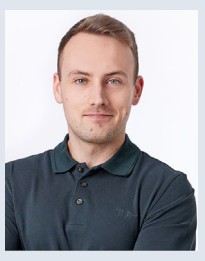

there is mounting evidence that adapted physical activity (APA) may attenuate loss of motor function and delay, or even reverse muscle wasting (Sacconi et al., 2015; Wang & Tawil, 2016). Some studies have reported improvements in aerobic capacity, walking speed and self-reported fatigue following exercise training (Andersen et al., 2015; Bankolé et al., 2016; Olsen et al., 2005; Voet et al., 2014), whereas others failed to report improvements in muscle mass and strength in response to high-intensity interval training and/or strength training (Andersen et al., 2017; van der Kooi et al., 2004). Notably, the impact of chronic exercise on degenerative events and regenerative capacity in skeletal muscle of FSHD is currently unknown.

The muscle stem cells, that isthe satellite cells, are heavily recruited over the course of a myopathy to ensure replacement of injured muscle fibres (Bankolé et al., 2013). The progressive muscle degeneration seen in myopathic patients may be underpinned by intrinsic satellite cell dysfunction. Indeed, cells isolated from FSHD muscle demonstrated impaired regenerative capacity and morphological defects when cultured *in vitro* (Barro et al., 2010). Although chronic exercise is known to expand the satellite cell pool in healthy skeletal muscle (Charifi et al., 2003; Kadi et al., 2004), it remains to be determined whether exercise may alter the satellite cell pool in slow and fast muscle fibres of FSHD patients. Moreover, another important aspect of the regenerative history of muscle cells is the telomere length (Kadi & Ponsot, 2010). Telomeres, which are repeated DNA sequences located at the end of the chromosomes, shorten with each round of cell division until reaching a critical length associated with impaired proliferative capacity and cellular senescence (Allsopp et al., 1992). In healthy adult muscle, chronic exercise does not impair the length of telomeres, although data on exercise-related changes in muscle telomere length of FSHD patients are lacking (Kadi et al., 2008; Osthus et al., 2012; Ponsot et al., 2008; Rae et al., 2010).

We have previously conducted a 24 week multicentre randomized controlled trial to determine the middle-term effects of exercise training on muscle function in FSHD. We reported significant improvements in physical function and muscle fibre cross-sectional area (fCSA) in patients with FSHD after exercise training (Bankolé et al., 2016). In the present study, we aimed to further explore the effects of the randomized controlled trial on degenerative events and muscle regenerative capacity, evaluated with satellite cell content and telomere length in skeletal muscle of FSHD patients.

## Methods

### Ethical approval

Before the commencement of the study, written informed consent was obtained from all patients. All procedures

**Table 1. Baseline participant characteristics**

|  | Control group | Training group |
|---|---|---|
| Patients | *n* = 8 (7M/1F) | *n* = 8 (5M/3F) |
| Age (years) | 44 ± 10 | 40 ± 13 |
| Body mass (kg) | 72 ± 11 | 73 ± 13 |
| Height (cm) | 177 ± 11 | 172 ± 7 |
| Body mass index | 23 ± 4 | 24 ± 4 |

Data are presented as the mean ± SD. F, female; M, male.

followed the ethical principles stated in the *Declaration of Helsinki* and were approved by the local ethical board (Comité de Protection des Personnes Sud-Est 1, France), number 2010-A00288-31. This trial is registered with ClinicalTrials.gov (number NCT01116570).

### Participants

Sixteen patients (12 males and four females) diagnosed with FSHD volunteered for this study, as described previously (Bankolé et al., 2016). Patients were eligible and included for participation based on the following criteria; genetically verified FSHD type I, ability to perform cycling exercise and age ≥18 years. Exclusion criteria were a history of cardiovascular disease, evidence of diabetes, inflammatory disease or a body mass index ≥35 kg m$^{-2}$. After inclusion, patients were randomized into a control group (Control, seven males/one female) or a training group (Training, five males/three females). Baseline characteristics of the patients are presented in Table 1.

### Exercise training program

The 24 week training program has been described previously (Bankolé et al., 2016). Briefly, patients performed home-based exercise training three times weekly on a stationary ergocycle (35 min per session). The program combined continuous aerobic training followed by strength training and high-intensity interval training. Two times weekly patients trained at 60% of maximal aerobic power (MAP) followed by five sets of 10 near-maximal revolutions, whereas one session per week was performed at 40% of MAP interspersed by five sprints corresponding to 80% of MAP (60 s per sprint). An APA teacher accompanied the first six sessions, on average. Then, one weekly session was supervised at home and two other sessions were supported via phone. The APA teacher monitored heart rate recordings and adjusted exercise intensities throughout the intervention. Patients included in the control group were instructed to maintain their daily activities during the study period.

## Muscle biopsy sampling

Muscle biopsies were collected from the vastus lateralis muscle before and after the intervention using the Weil–Blakesley percutaneous technique. All biopsies were taken 2–7 days after the last completed exercise session. Fibre bundles suited for immunohistochemistry were aligned in parallel, placed in optimal cutting temperature compound, and submerged in isopentane chilled by liquid nitrogen. Samples for telomere length analysis were immediately frozen in liquid nitrogen and stored at −80°C.

## Immunohistochemistry

Muscle biopsies were cut into 5-mm cross-sections at −21°C using a cryostat (CM1850; Leica, Wetzlar, Germany). Cryosections were mounted on glass slides and stored at −80°C until being used for subsequent immuno-histochemical analyses.

To perform a fibre type-specific satellite cell analysis, we adopted a previously described staining protocol (Mackey et al., 2010). Following fixation for 5 min with 2% paraformaldehyde (Histolab, Gothenburg, Sweden) sections were blocked (1% normal goat serum, 1% fat-free milk) for 30 min, and incubated overnight with a Pax7 antibody (199010; Abcam, Cambridge, UK; dilution 1:300). The next day, sections were washed 2 × 5 min in phosphate-buffered saline (PBS) and incubated with a biotinylated secondary antibody (BA-2000; Vector Laboratories, Newark, CA, USA; dilution 1:200), followed by Vectastain ABC reagent (PK6100; Vector Laboratories) and the DAB substrate kit (SK-4105; Vector Laboratories). Sections were then incubated for 60 min with three antibodies against laminin (D18; dilution 1:100), myosin heavy chain type I (BA-F8; dilution 1:300) and type IIA (SC-71; dilution 1:300) from Developmental Studies Hybridoma Bank (DSHB, Iowa City, IA, USA). After washing 2 × 5 min in PBS, fluorescent secondary antibodies were applied for 60 min (Alexa Fluor 488 IgG2A, cat. no. A-21131; Alexa Fluor 488 IgG2B, cat. no. A-21141; and Alexa Fluor 568 IgG1, cat. no. A-21124; all purchased from Invitrogen, Taastrup, Denmark). Finally, the slides were mounted with mounting media containing 4′,6-diamidino-2-phenylindole (DAPI) (Prolong Gold Antifade Reagent; Molecular Probes, Eugene, OR, USA). The staining rendered satellite cells in brown, nuclei in blue, slow muscle fibres and laminin in green, fast type IIA fibres in red, and fast type IIX were left unstained, as shown in Fig. 1.

As a marker of muscle regeneration, cross-sections were stained with an antibody against developmental myosin heavy chains (MyHC-d). After fixation and blocking, sections were incubated for 60 min with an antibody against MyHC-d (NCL-MHCd; Leica Biosystems, Newcastle, UK; dilution 1:100), together with laminin (D18; DSHB; dilution 1:20). After washing 2×5 min in PBS, sections were incubated for 30 min with secondary antibodies (Alexa Fluor 488 IgG2A and 568 IgG1; Invitrogen) followed by DAPI mounting.

To evaluate the presence of local muscle macrophages infiltrates, sections were blocked and incubated for

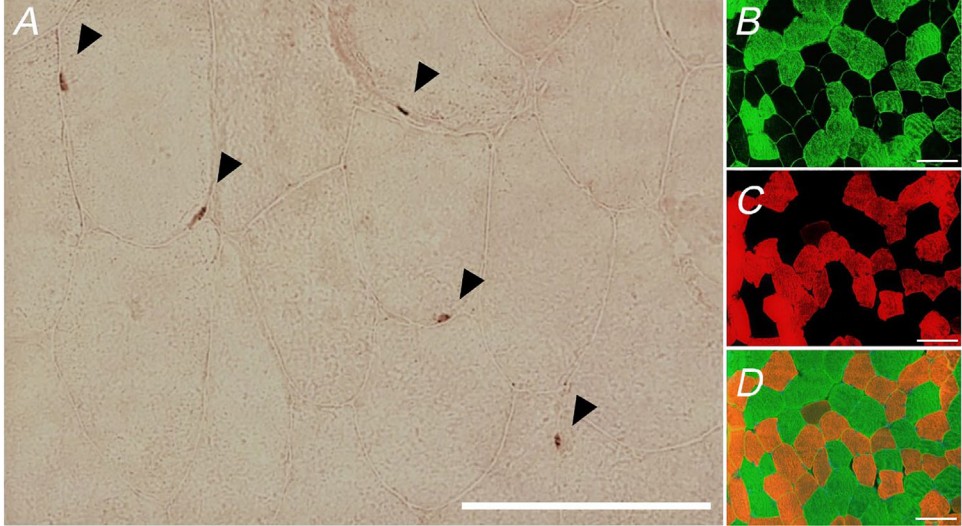

**Figure 1. Representative images of immunohistochemical identification of satellite cells, laminin, and slow and fast type muscle fibres**
*A*, satellite cells stained brown (indicated by arrowheads) using the Pax7-antibody and immunoenzymatic detection. *B*, fluorescence staining of slow-type muscle fibres and the basal membrane. *C*, fluorescence staining of fast type muscle fibres. *D*, merged fluorescence image used for quantification of fibre type composition. Scale bars = 100 µm. [Colour figure can be viewed at wileyonlinelibrary.com]

60 min with an antibody against cell surface marker of pan-macrophages (CD68; Dako Agilent Technologies; dilution 1:100) and laminin (D18; DSHB; dilution 1:20). Secondary antibodies were applied for 30 min (Alexa Fluor 488 IgG2A and 568 IgG1; Invitrogen), before being mounted with DAPI.

### Image acquisition and analysis

Stained sections were digitally captured with fluorescence microscopy (Nikon Eclipse E400; Nikon Instruments, Europe, Amstelveen, The Netherlands) with an attached digital camera (Insight Camera; SPOT Imaging Sterling Heights, MI, USA). Image processing and analysis were performed using ImageJ (National Institutes of Health, Bethesda, MD, USA).

Fibre type composition was determined in each section by counting the number of each fibre type divided by the total number of fibres. The measurements of fCSA included a minimum of 100 fibres per fibre type, with separate measurements for the three fibre types. Fast type IIX fibres, however, were excluded from the analysis as a result of low amounts. Fibres in the periphery of the section and fibres with freezing artefacts were not included. Fibre circularity was calculated using the previously defined formula (Charifi et al., 2003), and did not differ between pre-post biopsies (data not shown). In the present study, we present frequency histograms of fCSA as additional information to data previously shown (Bankolé et al., 2016). Satellite cells refer to $Pax7^+/DAPI^+$ cells located inside the laminin border. Once located using the light microscope, these $Pax7^+$ cells were marked on fluorescence images together with their associated fibre type. Satellite cell content was determined on the whole muscle section, with an mean $\pm$ SD of $636 \pm 291$ fibres being included for analysis. Nuclei were identified using the DAPI staining and were determined myonuclei if located inside the basal lamina. To reliably determine myonuclei content, 75 fibres per fibre type were assessed. The myonuclear domain (expressed in $\mu m^2$, was obtained by dividing the average fCSA by the average number of myonuclei per fibre.

For muscle regenerative signs, the number of fibres displaying immunoreactivity for MyHC-d and the number of fibres with centralized myonuclei were quantified and expressed as a proportion of the total fibre pool (%). Macrophages were defined as cells that stained positive for the CD68-antibody with a cell nucleus (DAPI) within the centre of the staining. Cells were not considered macrophages if the staining of the nucleus did not match in size or position with the CD68-staining. Macrophage infiltration was assessed by counting the total number of $CD68^+/DAPI^+$ puncta in a given window (six to eight images per cross-section) and expressed per 100 fibres.

The relative location of the macrophages (perimysial *vs.* endomysial site) was not quantitatively assessed. The mean $\pm$ SD number of fibres per cross-section included for analysis of $CD68^+/DAPI^+$ cells was $574 \pm 261$ fibres. Care was taken not to include fibres cut longitudinally. In addition, sections stained with haematoxylin and eosin were analysed to evaluate the possible presence of overt histopathological signs (with an example shown in Fig. 2).

As a result of issues with tissue freezing in one specimen, histological data were obtained from 15 patients (Control; $n = 7$, Training; $n = 8$). All analyses were performed by a single investigator who was blinded to the participant coding and the study design.

### Telomere length

Total genomic DNA was purified from muscle biopsies using a commercially available kit (Nucleospin Tissue DNA; Macherey-Nagel, Düren, Germany) in accordance with the manufacturer's instructions. DNA was later quantified using Nanodrop 2000c (Thermo Fisher Scientific, Waltham, MA, USA) and stored at $-20°C$. Relative muscle telomere length was measured using quantitative real-time PCR technique (Cawthon, 2002). Real-time PCR for telomere (T) and single copy gene expression (S) were separately performed using Rotor-Gene SYBR Green RT-PCR Master mix (Qiagen, Hilden, Germany) on a Rotor-Gene Q themocycler (Qiagen). The specificity of PCR product amplification was confirmed by melting curves and amplification efficiencies were validated by standard dilution series. The relative mean skeletal muscle telomere length was expressed as a T/S ratio using the $2^{-\Delta\Delta CT}$ method. Samples for telomere length were loaded in triplicates and 36B4 in duplicates. The primer sequences for telomere and 36B4 have been described previously (Brouilette et al., 2007). Muscle telomere length data includes 15 patients because of technical difficulties during the DNA purification phase

### Statistical analysis

Data are presented as the mean $\pm$ SD or means with individual data points. Baseline between-group comparisons were performed using the Mann–Whitney *U* test. The relative frequency of fibres in each fibre size category was compared using paired sample *t* tests within groups. Aligned rank transform (ART) two-way mixed ANOVA with one between-subject factor (group), one within-subject factor (time) and interaction terms (group $\times$ time) was used to investigate the effects of exercise training in all the variables. The original data first underwent ART (ARTool for Windows, version 2.1.2; https://depts.washington.edu/acelab/proj/art), and

then the ranked data were analysed with the usual ANOVA procedure (Elkin et al., 2021; Wobbrock et al., 2011). Least significant difference pairwise comparisons were conducted when significant interactions were found. Statistical analysis was performed using SPSS, version 28.0 (IBM Corp., Armonk, NY, USA). $P < 0.05$ was considered statistically significant.

## Results

Histopathological analysis of baseline muscle biopsies did not reveal signs of overt muscle degeneration, containing only a small proportion of fibres with centralized nuclei and MyHC-d (Table 2). Importantly, the training intervention did not increase the number of fibres with centralized myonuclei or fibres expressing MyHC-d (Table 2). A small subset of all biopsies displayed myofibre grouping. The presence of $CD68^+/DAPI^+$ cells was observed in samples from all patients. Based on visual inspection of the staining pattern, these cells were mainly located in the perimysial area (Fig. 2C), but this was highly variable among patients and some manifested high infiltration also in the endomysial area alongside signs of myophagocytosis (Fig. 2D). However, there were no significant changes in immune cell infiltration in response to the intervention as evaluated with the CD68-staining. Providing further support to our previous data showing muscle fibre hypertrophy as a result of the intervention (Bankolé et al., 2016), a clear rightward shift in the histogram distribution in each of the two fibre types could be observed in the Training group (Fig. 3). The proportion of fibres with CSAs between 3000–4000 $\mu m^2$ decreased in slow and fast fibres ($P = 0.022$ and $P = 0.020$, respectively) and the proportion of fibres with CSAs between 5000–6000 $\mu m^2$ increased only in fast fibres ($P = 0.008$) (Fig. 3C and D). The analysis of satellite cell content at baseline revealed a higher number of satellite cells associated with slow compared to fast muscle fibres ($P = 0.021$). After the intervention, a significant (+75 % increase; $P = 0.015$) expansion of the satellite cell pool occurred specifically in fast muscle fibres in the Training group (Fig. 4B). No significant changes in satellite cell content occurred in the Control group. We did not observe any significant changes in the number of myonuclei or in the myonuclear domain of either slow or fast muscle fibres in response to exercise training (Table 2). Analysis of telomere length revealed that the 24 week intervention was not associated with significant changes in telomere length in either of the two groups (Fig. 4C).

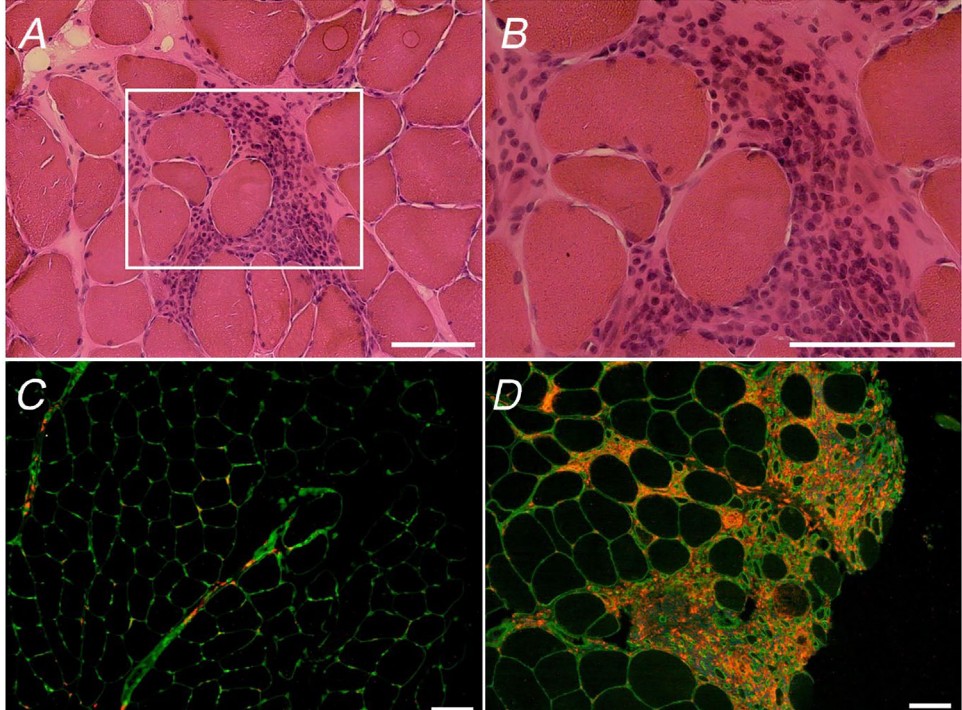

**Figure 2. Representative images of FSHD muscle cross-sections with immune cell infiltration**
*A*, haematoxylin and eosin stained cross-section showing typical morphological features of FSHD such as irregular fibre size and presence of inflammatory infiltrates. *B*, an area is seen at a higher magnification (40×). *C* and *D*, fluorescence labelled $CD68^+$ cells (red) and the basal membrane (green). Scale bars = 100 μm. [Colour figure can be viewed at wileyonlinelibrary.com]

**Table 2. Muscle fibre characteristics before and after 24 weeks of exercise training**

| | Control group | | Training group | |
|---|---|---|---|---|
| | T0 | T24 | T0 | T24 |
| **Fibre type composition (%)** | | | | |
| Type I | 25 ± 13* | 31 ± 11 | 40 ± 12 | 37 ± 18 |
| Type IIA | 68 ± 11* | 61 ± 6 | 52 ± 8 | 57 ± 16 |
| Type IIX | 4 ± 6 | 4 ± 5 | 4 ± 5 | 3 ± 4 |
| Intermediate fibres | 3 ± 4 | 4 ± 3 | 4 ± 5 | 3 ± 3 |
| **Myonuclei/fibre** | | | | |
| Type I | 2.82 ± 1.16 | 2.91 ± 1.11 | 2.23 ± 0.33 | 2.13 ± 0.54 |
| Type IIA | 2.64 ± 0.86 | 2.77 ± 0.99 | 2.32 ± 0.51 | 2.41 ± 0.65 |
| **Myonuclear domain ($\mu m^2$)** | | | | |
| Type I | 2956 ± 999 | 2503 ± 893 | 2410 ± 471 | 2688 ± 707 |
| Type IIA | 2426 ± 504 | 2330 ± 580 | 2264 ± 739 | 2272 ± 1046 |
| **Muscle regeneration (%)** | | | | |
| MyHC-d$^+$ fibres | 0.06 ± 0.17 | 0.36 ± 0.78 | 0.12 ± 0.21 | 0.17 ± 0.27 |
| Fibers with centralized nuclei | 2.57 ± 1.4 | 2.21 ± 2.0 | 1.59 ± 0.9 | 3.90 ± 4.8 |
| **Macrophages/100 fibres** | | | | |
| CD68$^+$/DAPI$^+$ cells | 16.4 ± 9.7 | 32.5 ± 27.3 | 18.2 ± 8.7 | 18.1 ± 12.6 |

Data are presented as the mean ± SD. Control; *n* = 7, Training; *n* = 8). MyHC-d; developmental myosin heavy chain. *Significant between-group difference at baseline.

## Discussion

The main finding of the present study was that exercise training in FSHD patients induced fibre type-specific expansion of the satellite cell pool but was not associated with alterations in skeletal muscle telomere length.

The ability of skeletal muscle to repair itself is largely reliant on muscle stem cells, whose activity ensures that daily wear and tear on muscle fibres is repaired and that new muscle fibres are formed following damaging insults (Lepper et al., 2011). However, during the course of a muscular dystrophy, these cells are often unable to counteract elevated rates of degeneration, resulting in the formation of tissue fibrosis, which ultimately causes gradual deterioration of muscle mass and function (Sacconi et al., 2015; Wang & Tawil, 2016). In the present study, satellite cell content was assessed in FSHD patients before and after exercise training. At baseline, satellite cell counts were similar to those previously described in muscle from healthy adults (Horwath, Moberg, et al., 2020; Kadi et al., 2006), indicating that the muscle stem cell pool is maintained in FSHD patients manifesting a mild dystrophic phenotype. However, fast muscle fibres displayed lower satellite cell counts than slow fibres, both at baseline and at the end of the intervention. This fibre type-specific pattern has been observed in other muscle diseases as well, such as in Duchenne muscular dystrophy (Bankolé et al., 2013), suggesting that reduced habitual activity levels, namely decreased neuronal input and/or less mechanical loading, may impair the stem cell pool, particularly in fast type muscle fibres, which are less regularly recruited during daily activities. It could also be speculated that a greater pool of satellite cells associated with slow fibres acts as a sparing mechanism, making these fibres more resilient to daily fibre lesions. Nonetheless, because we report a significant increase expansion of the satellite cell pool in response to exercise training, the myogenic response of fast muscle fibres appears to be preserved. This adaptation might prove important for maintaining a functional pool of fast-contracting muscle fibres in patients with FSHD.

Increased satellite cell counts following a period of muscle loading are well described in healthy adults across different ages (Kadi et al., 2004; Mackey et al., 2011). Here, we report for the first time that FSHD patients can expand the satellite cell pool in response to combined exercise training in their fast muscle fibres, providing evidence that the myogenic potential of FSHD muscle is intact. It should be noted that the training regimen also had a positive impact on satellite cells associated with slow muscle fibres (six out of eight patients displayed an increase); however, this did not reach statistical significance ($P$ = 0.069). Furthermore, given the importance of satellite cells for maintaining muscle homeostasis, this may represent a finding of clinical significance because an enriched cell pool may augment the capacity to repair and replace injured tissue. The increased abundance of satellite cells associated with fast fibres is particularly relevant considering that these fast fibres constituted a majority of the total fibre pool. However, whether these cells can delay or reverse muscle wasting during the progression of the disease remains unknown when considering that the

regenerative potential of FSHD muscle may be inhibited by other factors, such as an intrinsic cellular dysfunction or a chronically inflamed systemic environment in which these cells reside (Barro et al., 2010; Bosnakovski et al., 2008; Statland et al., 2014). Nevertheless, the present study provides evidence for exercise training as a possible strategy to mitigate muscle wasting in FSHD patients, complementing prior work showing favourable effects of exercise training on muscle strength, endurance and functional outcomes such as walking speed (Andersen et al., 2015; Olsen et al., 2005; Voet et al., 2014).

In the present study, myonuclear content remained unchanged in both fibre types despite increases in muscle fibre size, as reported in our previous study (Bankolé et al., 2016). Increased muscle loading through combined exercise training thereby provided a sufficient stimulus to renew and replenish the satellite cell pool but did not stimulate satellite cell fusion and subsequent accretion of myonuclei. Considering emerging research describing fusion independent roles of satellite

cells, the expanded satellite cell pool observed here may reach beyond the classical role of repairing tissue damage and donating nuclei (Murach et al., 2021). For example, it is shown that satellite cell-derived factors can modulate gene expression of proteins located in the extracellular matrix to create a more favourable environment for long-term adaptations (Murach et al., 2020). In the context of FSHD, a disease characterized by high rates of fat infiltration and fibrosis, increased satellite cell communication with the extracellular matrix may represent a candidate mechanism by which increased physical activity can attenuate or even reverse fibrotic deposition. Although these fusion independent roles of satellite cells are so far poorly understood, they represent an interesting avenue for future research.

Although the absence of myonuclear incorporation and the lack of changes to the myonuclear domain in the present study contrast with some previous findings (Petrella et al., 2008), it is important to consider that much of our understanding of this process is based on research

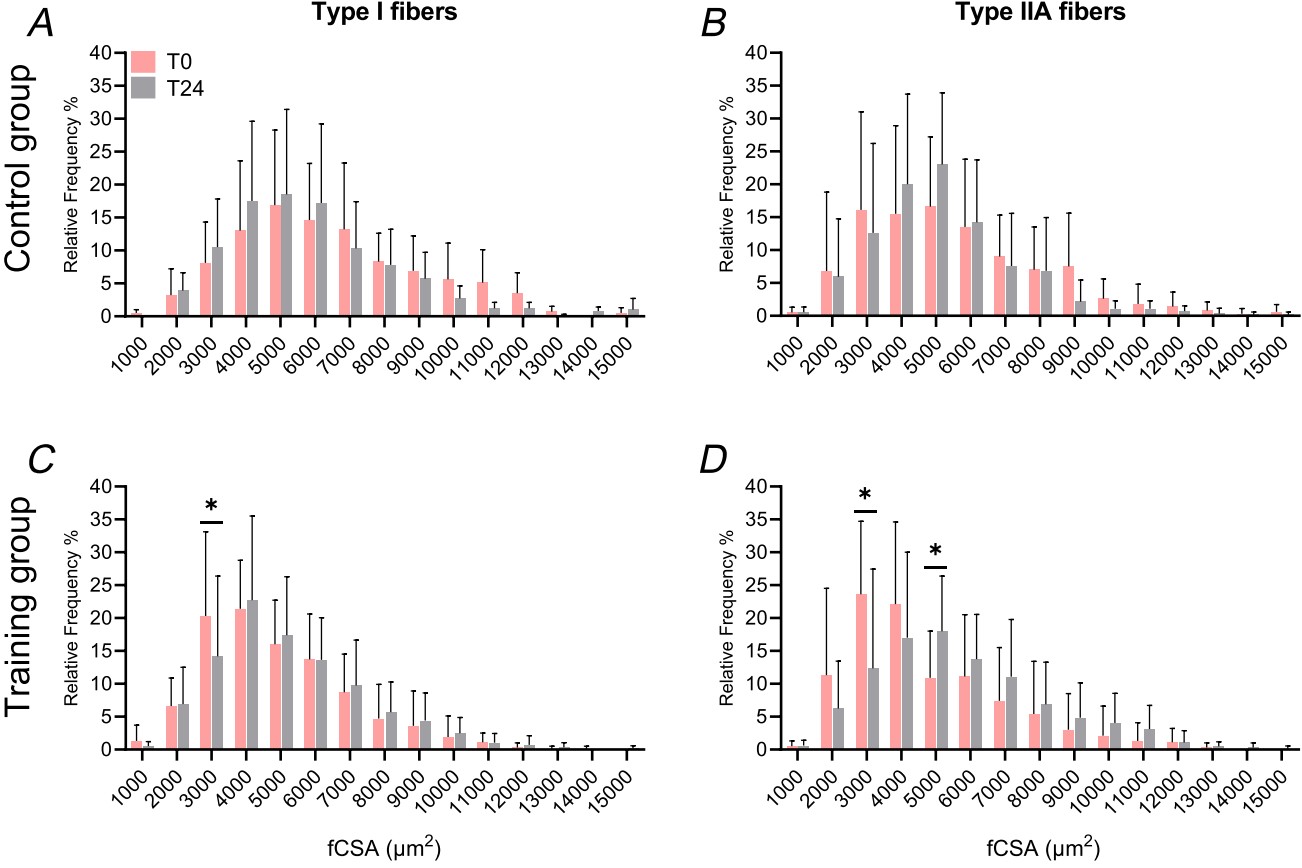

**Figure 3. Muscle fibre cross-sectional area before and after 24 weeks of exercise training presented as a frequency histogram**
*A*, slow type I muscle fibres in the Control group. *B*, fast type IIA muscle fibres in the Control group. *C*, slow type I muscle fibres in the Training group. *D*, fast type IIA muscle fibres in the Training group. Data are presented as the mean ± SD. Analysis of cross-sectional area contains data from 15 patients (Control group; *n* = 7, Training group; *n* = 8). *Significantly different from before. [Colour figure can be viewed at wileyonlinelibrary.com]

conducted in healthy adult tissue. The same principles are therefore not necessarily applicable to dystrophic muscle. For example, it is reasonable to assume that a cellular process such as myonuclear accretion is not prioritized, whereas the satellite cells are highly recruited for repair and regenerative purposes. It is also possible that our finding represents an inherent defect of the satellite cells to fuse with existing fibres in FSHD muscle. Despite this, our data further strengthens the notion that muscle growth can occur regardless of parallel increases in myonuclei number, and so also in muscles affected by FSHD (Horwath, Apró, et al., 2020; Kadi et al., 2004; Verney et al., 2008). Indeed, several studies of healthy volunteers have shown that moderate increases in fibre size can be sustained by the existing transcriptional machinery, without the need for additional myonuclei (Horwath, Apró, et al., 2020; Verney et al., 2008). By contrast, in light of the myonuclear ceiling theory (Kadi, 2000; Kadi et al., 2004), which proposes that additional myonuclei are required once the muscle fibre has surpassed a certain size, the lack of myonuclear accretion was unexpected given the low nuclei-to-cytoplasmic volume ratio observed in these samples, comprising a larger myonuclear domain than normally seen in healthy muscle tissue (Kadi et al., 2004). This supports the notion that a relatively larger myonuclear domain at baseline is not a driving factor for myonuclear accretion during exercise-induced muscle growth (Snijders et al., 2016).

In the present study, we noted that our patient cohort had a fast-glycolytic muscle profile (on average 68 % fast fibres). An increased proportion of fast fibres in FSHD muscle has not previously been observed because others have reported that FSHD muscle is either similar to healthy controls (Lassche et al., 2021) or shifted in the opposite direction, that is, fast-to-slow, in human and rodent muscle (Celegato et al., 2006; D'Antona et al., 2007; Hubregtse et al., 2024). The exact reason for

our observation is therefore not readily apparent, but it may reflect a shift towards a fast-glycolytic phenotype because of reduced habitual activity levels as a secondary consequence of the disease (Gallagher et al., 2005). Other possible reasons for this predominance of fast fibres may include selective degradation of slow fibres through insufficient regeneration or loss of fibres because of denervation, although this remains highly speculative at present and warrants future work. Furthermore, the training period was not accompanied by the typical fast-to-slow-fibre type transition, which is consistent with earlier work in FSHD patients (Olsen et al., 2005).

Concerns were previously raised about the possible negative influence of exercise-induced workload on the regenerative potential of skeletal muscle and the question of whether exercise can provoke a shortening of telomere length in skeletal muscle of FSHD patients has never been previously addressed. In accordance with previous observations from our laboratory obtained in healthy active adults (Kadi et al., 2008; Ponsot et al., 2008), in the present study, we show that chronic exercise is not associated with altered telomere length in muscle cells of FSHD patients. Our data thus reinforce that enhanced satellite cell recruitment, as a result of chronic exercise in the form of combined aerobic and high-intensity interval training, does not accelerate muscle cell turnover. This information is important for ongoing debates related to the role of exercise in patients with muscle diseases (Gianola et al., 2020). We and others have previously provided evidence showing that exercise is a safe and well-tolerated therapy in FSHD patients (Andersen et al., 2017; Bankolé et al., 2016; Olsen et al., 2005). The present study extends this view by demonstrating that chronic exercise does not compromise the regenerative reserve of muscle cells. In addition, it should be considered that regulation of telomere length *in vivo* is more complex than described in cell culture systems and that telomeres

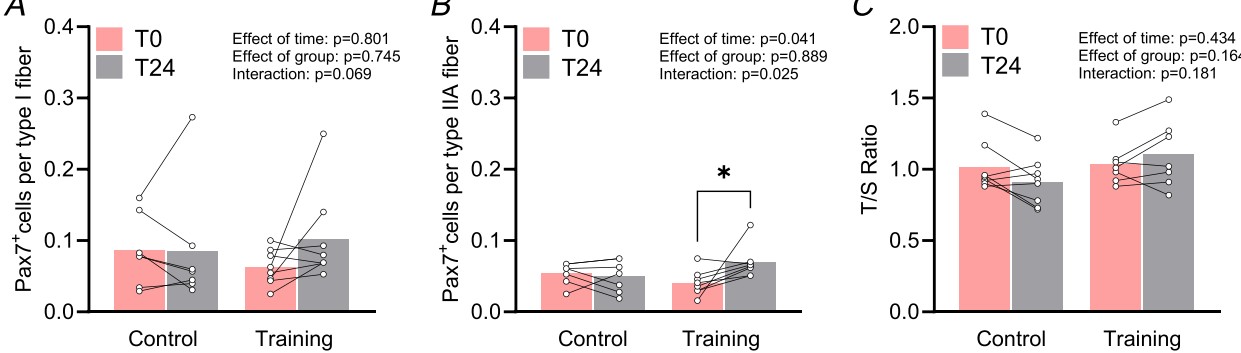

**Figure 4. Fibre type-specific satellite cell content and muscle telomere length before and after 24 weeks of exercise training**
Satellite cell content is reported for slow type I fibres (*A*) and fast type IIA fibres (*B*). Telomere length (*C*) is expressed as T/S ratio. Data are presented as means and individual values. Data on satellite cell content and telomere length contains data from 15 patients (Control group; *n* = 7, Training group; *n* = 8). *Significantly different from before. [Colour figure can be viewed at wileyonlinelibrary.com]

in muscle cells are dynamic structures, not exclusively influenced by the number of cell divisions, but also controlled by a network of different regulatory factors, such as telomerase and tankyrase 1, which we did not consider here (Ponsot et al., 2012; Werner et al., 2008, 2019). Because exercise is shown to modulate the expression of such factors in circulating blood cells (Werner et al., 2008, 2019), it is plausible that a similar mechanism contributed to our findings in skeletal muscle. Moreover, based on earlier observations in myopathic muscle (Ponsot et al., 2012), we cannot exclude the possibility that a telomere-protective pathway was active in these patients already prior to the intervention. Future mechanistic studies should thus address whether factors associated with telomere maintenance are differently regulated during exercise in healthy and diseased muscle tissue. In addition, because our measurements of telomere length were performed on whole muscle homogenates, we cannot exclude the involvement of other muscle resident cell types. Nevertheless, we still consider that our samples allowed for a fair comparison given that the rate of infiltration by other cell types remained constant over the course of the intervention.

In the present study, we observed significant inflammatory infiltrates, a cellular hallmark of the FSHD phenotype (Arahata et al., 1995), although we also showed that chronic exercise did not accentuate this muscle infiltration. A transient immune cell response after a damaging event is critical for the onset of muscle regenerative processes, whereas chronic inflammation, manifested in certain myopathies, impairs satellite cell activity and leads to the formation of tissue fibrosis (Chang et al., 2016). In this regard, the positive effect on satellite cells observed might have been mediated by the muscle inflammatory status as a result of a shift between pro-inflammatory (M1) and anti-inflammatory (M2) macrophages. Indeed, evidence in myositis patients supports the notion that exercise training reduces local inflammation and thereby potentially creates a more permissive cellular environment for satellite cell activity and muscle adaptability (Nader et al., 2010). At present, this reasoning remains speculative and cannot be concluded from our data on macrophages expressing the CD68-antigen, which, in human tissue, is a known marker of pan-macrophages (Kosmac et al., 2018). We acknowledge that a more comprehensive analysis of the different subtypes of macrophages would have provided insights into the possible immunomodulatory role of exercise training in these patients. Nonetheless, given the association between infiltration of CD68[+] cells and muscle injury (Bernard et al., 2022), our analysis of macrophage abundance provided relevant information regarding ongoing muscle regeneration, which, in the present study, was not exacerbated as a function of the intervention. Lastly, our patient cohort generally manifested mild signs of ongoing muscle regeneration in the vastus lateralis muscle, as evaluated by muscle fibres expressing MyHC-d and centralized myonuclei. The regenerative response in FSHD biopsies has previously been shown to correlate well with the severity of pathology (Banerji et al., 2020). Importantly, these cellular markers did not change over the course of the intervention, indicating that the rate of muscle fibre degeneration was not altered in muscles of active patients.

The present study provides novel information regarding the myogenic response to middle-term training in skeletal muscle of FSHD patients. An expansion of the satellite cell pool in fast muscle fibres occurred in parallel with muscle fibre hypertrophy. Our findings suggest that combined exercise training elicits beneficial muscular adaptations without impairing skeletal muscle regenerative capacity (telomere length) in patients with FSHD.

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

## Additional information

### Data availability statement

The data that support the findings of the present study are available from the corresponding author upon reasonable request.

### Competing interests

The authors declare that they have no competing interests.

### Author contributions

L.F. and F.K. designed the study. OH and DMR performed the experiments at Örebro University under the supervision of E.P. and F.K.O.H. and D.M.R. analysed the data and interpreted

the data together with E.P., L.F. and F.K. O.H. drafted a first version of the manuscript. All authors revised the manuscript critically for important intellectual content. All authors have read and approved the final version of the manuscript submitted for publication and agree to be accountable for all aspects of the work in ensuring that questions related to the accuracy or integrity of any part of the work are appropriately investigated and resolved. All persons designated as authors qualify for authorship, and all those who qualify for authorship are listed.

## Funding

This study was supported by the Association Française contre la Myopathie (AFM). The funding source had no role in study design, data collection, data analysis, data interpretation or writing of the report.

## Acknowledgments

We warmly thank the patients for their participation.

## Keywords

facioscapulohumeral muscular dystrophy, muscle fibre, muscle regeneration, myogenesis, Pax7

## Supporting information

Additional supporting information can be found online in the Supporting Information section at the end of the HTML view of the article. Supporting information files available:

**Peer Review History**

