## [Peer Review History · The Journal of Physiology]

Increased muscle satellite cell content and preserved telomere length in response to combined exercise training in patients with FSHD

Oscar Horwath, Diego Montiel Rojas, Elodie Ponsot, Léonard Féasson, and Fawzi Kadi
DOI: 10.1113/JP287033

Corresponding author(s): Fawzi Kadi (fawzi.kadi@oru.se)

Review Timeline:

Submission Date:	31-May-2024
Editorial Decision:	05-Jul-2024
Revision Received:	12-Dec-2024
Editorial Decision:	09-Jan-2025
Revision Received:	13-Jan-2025
Accepted:	17-Jan-2025

Senior Editor: Paul Greenhaff

Reviewing Editor: Matthew Brook

Transaction Report:

Dear Dr Kadi,

Re: JP-RP-2024-287033 "Increased muscle satellite cell content and preserved telomere length in response to combined exercise training in patients with FSHD" by Oscar Horwath, Diego Montiel Rojas, Elodie Ponsot, Léonard Féasson, and Fawzi Kadi

Thank you for submitting your manuscript to The Journal of Physiology. It has been assessed by a Reviewing Editor and by 2 expert referee and we are pleased to tell you that it is potentially acceptable for publication following satisfactory major revision.

LANGUAGE EDITING AND SUPPORT FOR PUBLICATION: If you would like help with English language editing, or other article preparation support, Wiley Editing Services offers expert help, including English Language Editing, as well as translation, manuscript formatting, and figure formatting at www.wileyauthors.com/eoo/preparation. You can also find resources for Preparing Your Article for general guidance about writing and preparing your manuscript at www.wileyauthors.com/eoo/prepresources.

REVISION CHECKLIST:

We look forward to receiving your revised submission.

Yours sincerely,

Paul Greenhaff
Senior Editor
The Journal of Physiology

REQUIRED ITEMS FOR REVISION

- Author photo and profile. First or joint first authors are asked to provide a short biography (no more than 100 words for one author or 150 words in total for joint first authors) and a portrait photograph. These should be uploaded and clearly labelled together in a Word document with the revised version of the manuscript. See Information for Authors for further details.

- Your manuscript must include a complete Additional Information section, including competing interests; funding; author contributions and acknowledgements.

- Please ensure that the Article File you upload is a Word file.

- Papers must comply with the Statistics Policy: https://jp.msubmit.net/cgi-bin/main.plex?form_type=display_requirements#statistics.

In summary:

- If n {less than or equal to} 30, all data points must be plotted in the figure in a way that reveals their range and distribution. A bar graph with data points overlaid, a box and whisker plot or a violin plot (preferably with data points included) are acceptable formats.

- If $n > 30$, then the entire raw dataset must be made available either as supporting information, or hosted on a not-for-profit repository, e.g. FigShare, with access details provided in the manuscript.

- 'n' clearly defined (e.g. x cells from y slices in z animals) in the Methods. Authors should be mindful of pseudoreplication.

- All relevant 'n' values must be clearly stated in the main text, figures and tables.

- The most appropriate summary statistic (e.g. mean or median and standard deviation) must be used. Standard Error of the Mean (SEM) alone is not permitted.

- Exact p values must be stated. Authors must not use 'greater than' or 'less than'. Exact p values must be stated to three significant figures even when 'no statistical significance' is claimed.

- A Data Availability Statement is required for all papers reporting original data. This must be in the Additional Information section of the manuscript itself. It must have the paragraph heading 'Data Availability Statement'. All data supporting the results in the paper must be either: in the paper itself; uploaded as Supporting Information for Online Publication; or archived in an appropriate public repository. The statement needs to describe the availability or the absence of shared data. Authors must include in their statement: a link to the repository they have used, or a statement that it is available as Supporting Information; reference the data in the appropriate section(s) of their manuscript; and cite the data they have shared in the References section. Whenever possible, the scripts and other artefacts used to generate the analyses presented in the

paper should also be publicly archived. If sharing data compromises ethical standards or legal requirements then authors are not expected to share it, but must note this in their statement. For more information, see our Statistics Policy.

- Please include an Abstract Figure file, as well as the Figure Legend text within the main article file. The Abstract Figure is a piece of artwork designed to give readers an immediate understanding of the research and should summarise the main conclusions. If possible, the image should be easily 'readable' from left to right or top to bottom. It should show the physiological relevance of the manuscript so readers can assess the importance and content of its findings. Abstract Figures should not merely recapitulate other figures in the manuscript. Please try to keep the diagram as simple as possible and without superfluous information that may distract from the main conclusion(s). Abstract Figures must be provided by authors no later than the revised manuscript stage and should be uploaded as a separate file during online submission labelled as File Type 'Abstract Figure'. Please also ensure that you include the figure legend in the main article file. All Abstract Figures should be created using BioRender. Authors should use The Journal's premium BioRender account to export high-resolution images. Details on how to use and access the premium account are included as part of this email.

- Please include a full title page as part of your main article (Word) file, which should contain the following: title, authors, affiliations, corresponding author name and contact details, keywords, and running title.

EDITOR COMMENTS

Reviewing Editor:

Thank you for submitting to The Journal of Physiology. Your manuscript has been reviewed by two expert referees. The referees have highlighted that the authors have presented novel results using valuable samples from a patient population. However, the referees have highlighted further information is required in the manuscript. This includes, further methodological details, consistency with the previous Bankole 2016 publication, and further discussion of key findings.

Senior Editor:

Thank you for the manuscript submission to The Journal of Physiology (TJP), which has been considered by a reviewing editor and two expert reviewers. All are in agreement that the subject area that the authors have addressed is important given it focuses on a rare human disease. However, a number of major concerns have been raised which the authors need to address in a comprehensive manner if the manuscript is to be considered suitable for further consideration. This includes greater detail of laboratory and statistical analyses being provided, moderated of parts of the Discussion or adding further analyses (theoretical calculations and additional immunohistochemical analyses) to support points being made, and greater focus in the Discussion on the end point measures specific to this study. Importantly, the authors need to differentiate the scientific impact of this study from (Bankole et al. 2016) to avoid the findings being viewed as incremental in nature, which detracts from the scientific impact of the work. Finally, to meet TJP ethical guidelines, the approval number of the local ethical board (Comité de Protection des Personnes Sud-Est 1, France) needs to be included.

REFEREE COMMENTS

Referee #1:

The authors have completed additional analysis on a previously published cohort (Bankole et al. 2016) of FSHD type I patients who completed a 24 week exercise training program, they have quantified satellite cell content, markers of regeneration (including inflammatory cell infiltration) and telomere length before and after training. The results presented here are novel, especially due to the use of a FSHD patient population.

Results from this study, as highlighted by the authors, are from samples collected in a previously published study (Bankole et al. 2016), minimal new analysis is included in the current manuscript. To improve clarity, further information regarding the analysis of the histological sections should be included, for example how the perimysial area was confirmed and detail on how myophagocytosis was determined should be included. Further clarification on CD68 staining and analysis is needed, clearly indicating a positive cell on the representative image would improve clarity for the reader. How statistical analysis regarding 'small' and 'large' CSA of muscle also fibres isn't clear.

The discussion could be improved by being more focused to the data presented here.

Referee #2:

While the presence of a healthy control group would have strengthened the design, the overall design of this study is sound and the major value of the current manuscript is that it represents human muscle tissue samples from a rare patient population. While the research is original and consists of robust data, some of the statements in the discussion need further support.

METHODS

Provide catalogue numbers of the Alex Fluor secondary antibodies (there are different types with the fluorophores used).

RESULTS

In Figures and Tables, T0 and T6 are used - define T6. In the Bankole 2016 paper T6 refers to six weeks of training, not the 24 implied in the current manuscript. It does not even seem that biopsies were collected at T6 according to the Bankole 2016 description of the study.

Table 2 lacks units for all parameters apart from fibre type composition so some of the data are impossible to interpret (per fibre/area, % immunoreactivity...?)

Figure 1: check scale bar - these would be extremely large muscle fibres (30,000 square microns) at the current scale, unless this patient happens to have unusually large fibres in the region displayed. According to your histograms, very few fibres were this size and the means reported in the Bankole study are 5300 - 7000 square microns.

Also add a scale bar to one of the fluorescent images since these are clearly different to the DAB image.

DISCUSSION

Satellite cells

The discussion includes appropriate reference to the fusion dependent roles of satellite cells but the more recent emerging fusion-independent roles of satellite cells (<https://pubmed.ncbi.nlm.nih.gov/34480776/>) could also be considered as these may be especially important in such a patient population where myonuclear incorporation occurs at a low rate. Particularly in light of the lack of myonuclear incorporation, I missed a discussion on the potential roles of the expanded satellite cell pool. What is the signal for satellite cells to proliferate, and their purpose, if they do not end up as new myonuclei.

"given the low nuclei-to-cytoplasmic volume ratio observed in these samples" - please provide the data or myonuclear domain (MND) size. Even though you don't have single fibre MNDs, you could calculate MND from the mean number of myonuclei/fibre and fCSA per biopsy data as an estimate.

CD68

"the presence of CD68+ cells was observed in samples from all patients, mainly located in the perimysial area, associated or not with myophagocytosis, as seen in Figure 2." And in the discussion: "In the present study, we observed significant inflammatory infiltrates in the perimysial site" - was this quantified, i.e. the number of CD68+ cells in perimysium vs endomysium? Please provide representative images that support your statement. It can be argued that Figure 2 shows just as much infiltration of inflammatory cells in the endomysium as the perimysium, where most of the cells seem to be

concentrated around a single muscle fibre which happens to be at the edge of a fascicle and therefore includes both endo- and peri-mysial areas surrounding its border.

Fibre types

It is curious that a higher proportion of IIX fibres was not detected at baseline. 3-4% IIX fibres is low in relation to the reports on untrained individuals in the literature. Furthermore, putting aside the higher proportion of type I fibres in the training group (40% vs 25% in the control group) at baseline, it seems in general that this patient group is characterised by a predominance of type II fibres. Could the authors speculate as to whether this might be genetic or due to a preferential loss of type I myofibres, potentially through denervation. An immunohistochemical assessment of denervation would be valuable in this context. Furthermore, the predominance of type II fibres makes the lower satellite cell content of type II versus type I fibres at baseline even more pertinent to the overall capacity for the muscle to maintain homeostasis with regard to regulation of fibrosis and signalling to the myofibre itself via satellite cell fusion-independent mechanisms (<https://pubmed.ncbi.nlm.nih.gov/34480776/>).

Telomere length

I appreciate that much of our understanding of telomere length is based on cell culture and does not take into account telomere length protective mechanisms that may act in vivo. Nonetheless perhaps the authors could calculate how much of a change in satellite cell telomere length would be required in order to detect a significant change when the analysis is performed on bulk muscle tissue as in the current manuscript. Or, alternatively, how many cell divisions (or number of dividing cells) would be required to reveal a detectable change in telomere length (theoretically). To warrant the current discussion on lack of change in telomere length (and the statement "The current study extends this view by demonstrating that chronic exercise does not compromise the regenerative reserve of muscle cells"), the reader needs to be able to assess the sensitivity of the method and whether there was sufficient statistical power to be able to detect a change in telomere length.

END OF COMMENTS

Response to reviewers – Horwath et al. “Increased muscle satellite cell content and preserved telomere length in response to combined exercise training in patients with FSHD”

Reviewing Editor:

Thank you for submitting to The Journal of Physiology. Your manuscript has been reviewed by two expert referees. The referees have highlighted that the authors have presented novel results using valuable samples from a patient population. However, the referees have highlighted further information is required in the manuscript. This includes, further methodological details, consistency with the previous Bankole 2016 publication, and further discussion of key findings.

Senior Editor:

Thank you for the manuscript submission to The Journal of Physiology (TJP), which has been considered by a reviewing editor and two expert reviewers. All are in agreement that the subject area that the authors have addressed is important given it focuses on a rare human disease. However, a number of major concerns have been raised which the authors need to address in a comprehensive manner if the manuscript is to be considered suitable for further consideration. This includes greater detail of laboratory and statistical analyses being provided, moderated of parts of the Discussion or adding further analyses (theoretical calculations and additional immunohistochemical analyses) to support points being made, and greater focus in the Discussion on the end point measures specific to this study. Importantly, the authors need to differentiate the scientific impact of this study from (Bankole et al. 2016) to avoid the findings being viewed as incremental in nature, which detracts from the scientific impact of the work. Finally, to meet TJP ethical guidelines, the approval number of the local ethical board (Comité de Protection des Personnes Sud-Est 1, France) needs to be included.

We thank the editors for the opportunity to revise our manuscript. We have addressed the specific points raised by the reviewers (see text below) and the approval number of the local ethical board has been added to the manuscript.

Reviewer 1:

The authors have completed additional analysis on a previously published cohort (Bankole et al. 2016) of FSHD type I patients who completed a 24 week exercise training program, they have quantified satellite cell content, markers of regeneration (including inflammatory cell infiltration) and telomere length before and after training. The results presented here are novel, especially due to the use of a FSHD patient population.

Results from this study, as highlighted by the authors, are from samples collected in a previously published study (Bankole et al. 2016), minimal new analysis is included in the current manuscript. To improve clarity, further information regarding the analysis of the histological sections should be included, for example how the perimysial area was confirmed and detail on how myophagocytosis was determined should be included. Further clarification on CD68 staining and analysis is needed, clearly indicating a positive cell on the representative image would improve clarity for the reader. How statistical analysis regarding 'small' and 'large' CSA of muscle also fibres isn't clear.

We thank the reviewer for the comments which have undoubtedly improved the manuscript.

To improve clarity for the reader, we have now added more details about the staining and subsequent quantification of the macrophages to the method section. For example, we point out that only cells positive for the CD68-antibody were considered macrophages if they co-localized with the DAPI staining, i.e., indicating a nucleus. These procedures are in line with previous studies evaluating the presence of macrophages in muscle tissue (PMID: 30148186).

However, in the present study, we did not quantify the specific location of the macrophages (perimysial vs endomysial site), nor did we assess if these cells were associated or not with myophagocytosis. We have therefore rephrased this accordingly throughout the manuscript and toned down any references to the specific location of the macrophages within the tissue, as well as stated this explicitly in the method section.

Despite that we did not perform a quantitative analysis, we observed a specific pattern when evaluating the CD68 staining which revealed that most of the macrophages were indeed located in the perimysial area (between fiber bundles) rather than in the endomysial site (between individual fibers). We have therefore reported this in the result section, nonetheless, to avoid any confusion, we clearly state that this is based on a visual inspection of the CD68-staining pattern and not based on any quantitative measures. To provide further support for this notion, we have added another representative image of the CD68-staining in Figure 2C.

The relative frequency of fibers in each fiber size category pre- and post-intervention was compared with paired sample t-tests. We have put this information in the statistical analyses section. However, to avoid confusion, we have removed the wording “small” and “large” fiber CSA and replaced it with the appropriate ranges of fiber CSA instead, for example, 5000-6000 μm^2 .

The discussion could be improved by being more focused to the data presented here.

We believe that the current version of the discussion is more focused on the data presented here. After revision, we have expanded the discussion on fiber type composition, the possible role of the expanded satellite cell pool as well as the lack of myonuclear accretion. However, if this is not satisfactory, maybe the reviewer could point us in the right direction and provide a more specific recommendation on which part of the discussion to further expand upon.

Reviewer 2:

While the presence of a healthy control group would have strengthened the design, the overall design of this study is sound and the major value of the current manuscript is that it represents human muscle tissue samples from a rare patient population. While the research is original and consists of robust data, some of the statements in the discussion need further support.

We thank the reviewer for comments and feedback on the current manuscript. We have addressed these concerns in a point-by-point fashion in the text below.

METHODS

Provide catalogue numbers of the Alex Fluor secondary antibodies (there are different types with the fluorophores used).

We appreciate the reviewer's concern in this matter as we are aware of this potential issue. We have now added the appropriate catalog numbers to the Alexa Fluor antibodies in the method section of the manuscript.

RESULTS

In Figures and Tables, T0 and T6 are used - define T6. In the Bankole 2016 paper T6 refers to six weeks of training, not the 24 implied in the current manuscript. It does not even seem that biopsies were collected at T6 according to the Bankole 2016 description of the study.

We apologize for the confusion that this might have created. We have now replaced T6 with T24 in all figures and tables throughout the manuscript. T0 and T24 represent the pre-and post-intervention timepoints, respectively.

Table 2 lacks units for all parameters apart from fibre type composition so some of the data are impossible to interpret (per fibre/area, % immunoreactivity...?)

We have now added appropriate units to Table 2.

Figure 1: check scale bar - these would be extremely large muscle fibres (30,000 square microns) at the current scale, unless this patient happens to have unusually large fibres in the region displayed. According to your histograms, very few fibres were this size and the means reported in the Bankole study are 5300 - 7000 square microns. Also add a scale bar to one of the fluorescent images since these are clearly different to the DAB image.

Appropriate scale bars have now been added to all images in Figure 1 and 2. We thank the reviewer for pointing this out to us.

DISCUSSION

Satellite cells

The discussion includes appropriate reference to the fusion dependent roles of satellite cells but the more recent emerging fusion-independent roles of satellite cells (<https://pubmed.ncbi.nlm.nih.gov/34480776/>) could also be considered as these may be especially important in such a patient population where myonuclear incorporation occurs at a low rate. Particularly in light of the lack of myonuclear incorporation, I missed a discussion on the potential roles of the expanded satellite cell pool. What is the signal for satellite cells to proliferate, and their purpose, if they do not end up as new myonuclei.

"given the low nuclei-to-cytoplasmic volume ratio observed in these samples" - please provide the data or myonuclear domain (MND) size. Even though you don't have single fibre MNDs, you could calculate MND from the mean number of myonuclei/fibre and fCSA per biopsy data as an estimate.

We agree with the reviewer that the newly discovered fusion-independent role of satellite cells is indeed intriguing. Unfortunately, there is little information on this topic in general and particularly in human subjects with muscle dystrophies. Despite this, we have included a short discussion on the increased satellite cell abundance observed in the present study and how it may contribute to muscle remodeling in patients with FSHD.

The reason for the lack of myonuclear incorporation in the present study is not entirely clear. However, it must be pointed out that much of our understanding of myonuclear addition in the context of muscle growth comes from work performed in healthy adults and may therefore not apply to muscles affected by muscular dystrophies. Few (if any) studies have previously reported muscle hypertrophy alongside satellite cell pool expansion in response to exercise training dystrophic patients which makes this a novel finding. Nonetheless, we have expanded on our discussion on this matter, and we speculate that this may reflect a cellular event with lower priority whilst the muscle is undergoing constant repair and remodeling.

Furthermore, as suggested by the reviewer, an estimation of the myonuclear domain, calculated as the mean number of myonuclei per fiber area, has been added to Table 2 in the manuscript. In the discussion, we have also clarified that a low nuclei-to-cytoplasmic volume ratio reflects a myonuclear domain that is larger than normally observed in healthy adult muscle tissue and provided a reference for this statement.

CD68

"the presence of CD68+ cells was observed in samples from all patients, mainly located in the perimysial area, associated or not with myophagocytosis, as seen in Figure 2." And in the discussion: "In the present study, we observed significant inflammatory infiltrates in the perimysial site" - was this quantified, i.e. the number of CD68+ cells in perimysium vs endomysium? Please provide representative images that support your statement. It can be argued that Figure 2 shows just as much infiltration of inflammatory cells in the endomysium as the perimysium, where most of the cells seem to be concentrated around a single muscle fibre which happens to be at the edge of a fascicle and therefore includes both endo- and peri-mysial areas surrounding its border.

This comment was raised also by reviewer 1 and we have addressed this in detail earlier in this document. In short, we did not quantify the specific location of the macrophages in this study. We have therefore toned down any reference to their specific location throughout the manuscript. However, in the result section, we mention that, in most cases, the staining pattern of the CD68 antibody was concentrated to the perimysial site, but in certain cases, there was macrophage infiltration also in the endomysial site. To strengthen this notion, we have added another representative image.

Fibre types

It is curious that a higher proportion of IIX fibres was not detected at baseline. 3-4% IIX fibres is low in relation to the reports on untrained individuals in the literature. Furthermore, putting aside the higher proportion of type I fibres in the training group (40% vs 25% in the control group) at baseline, it seems in general that this patient group is characterised by a predominance of type II fibres. Could the authors speculate as to whether this might be genetic or due to a preferential loss of type I myofibres, potentially through denervation. An immunohistochemical assessment of denervation would be valuable in this context. Furthermore, the predominance of type II fibres makes the lower satellite cell content of type II versus type I fibres at baseline even more pertinent to the overall capacity for the muscle to maintain homeostasis with regard to regulation of fibrosis and signalling to the myofibre itself via satellite cell fusion-independent mechanisms (<https://pubmed.ncbi.nlm.nih.gov/34480776/>).

We thank the reviewer for this insightful comment. We agree that the lower proportion of IIX fibers was surprising considering the high proportion of type II fibers in total and the seemingly “untrained” status of the patients. The exact reason for the higher proportion of type IIa fibers in our cohort is not known and we can therefore only speculate on this matter. We believe that this may occur mainly as a function of a relatively inactive lifestyle in these patients, as some studies have shown that prolonged inactivity, i.e., bed rest, is associated with a shift towards a greater proportion of type II fibers (PMID: 16128698). Also, the inclusion of specific markers of denervation, such as NCAM, would be interesting and may have revealed further clues about the origin of the fiber type composition, but as there were no overt signs of denervation in the biopsy samples, e.g., small and angulated fibers and fiber type grouping, we have not included such analyses in the present manuscript. Nonetheless, we have included a short discussion regarding the muscle fiber type composition in the manuscript where we cover some of these potential mechanisms. We have also, as the reviewer has pointed out, further highlighted the fact that increasing the satellite cell pool specifically in fast fibers may be of clinical importance due to the high abundance of fast fibers in the tissue (see discussion).

Telomere length

I appreciate that much of our understanding of telomere length is based on cell culture and does not take into account telomere length protective mechanisms that may act in vivo. Nonetheless perhaps the authors could calculate how much of a change in satellite cell telomere length would be required in order to detect a significant change when the analysis is performed on bulk muscle tissue as in the current manuscript. Or, alternatively, how many cell divisions (or number of dividing cells) would be required to reveal a detectable change in telomere length (theoretically). To warrant the current discussion on lack of change in telomere length (and the statement "The current study extends this view by demonstrating that chronic exercise does not compromise the regenerative reserve of muscle cells"), the reader needs to be able to assess the sensitivity of the method and whether there was sufficient statistical power to be able to detect a change in telomere length.

We appreciate the reviewer’s insightful comments on this important point. In response, we have addressed these limitations in the discussion section, where we acknowledge the advantages and limitations of using both PCR and bulk muscle tissue in our analysis. While it is challenging to assess sensitivity precisely as the reviewer suggests, we believe there is sufficient evidence to support our conclusion of telomere length stability in this study. First, the PCR-based method for telomere measurement is well-validated and widely used as an indicator of regenerative capacity,

even in bulk muscle tissue. In the specific case of FSHD, to answer previously raised concerns, if regular exercise had been an aggravating factor in terms of degenerative processes or impaired muscle regeneration, as it would expect a strong impact with changes to be substantial and detectable, even within the known methodological constraints. Moreover, the stability of all assessed indicators throughout the 24-week exercise program strengthens our conclusion regarding telomere length stability and supports our broader finding on the safety and potential benefits of exercise in this patient population.

Dear Dr Kadi,

Re: JP-RP-2024-287033R1 "Increased muscle satellite cell content and preserved telomere length in response to combined exercise training in patients with FSHD" by Oscar Horwath, Diego Montiel Rojas, Elodie Ponsot, Léonard Féasson, and Fawzi Kadi

Thank you for submitting your revised Research Article to The Journal of Physiology. It has been assessed by the original Reviewing Editor and Referees and has been well received. Some final revisions have been requested.

REVISION CHECKLIST:

We look forward to receiving your revised submission.

Yours sincerely,

Paul Greenhaff
Senior Editor
The Journal of Physiology

REQUIRED ITEMS

- Papers must comply with the Statistics Policy: https://jp.msubmit.net/cgi-bin/main.plex?form_type=display_requirements#statistics.

In summary:

- If n {less than or equal to} 30, all data points must be plotted in the figure in a way that reveals their range and distribution. A bar graph with data points overlaid, a box and whisker plot or a violin plot (preferably with data points included) are acceptable formats.
- If $n > 30$, then the entire raw dataset must be made available either as supporting information, or hosted on a not-for-profit repository, e.g. FigShare, with access details provided in the manuscript.
- 'n' clearly defined (e.g. x cells from y slices in z animals) in the Methods. Authors should be mindful of pseudoreplication.
- All relevant 'n' values must be clearly stated in the main text, figures and tables.
- The most appropriate summary statistic (e.g. mean or median and standard deviation) must be used. Standard Error of the Mean (SEM) alone is not permitted.
- Exact p values must be stated. Authors must not use 'greater than' or 'less than'. Exact p values must be stated to three significant figures even when 'no statistical significance' is claimed.

- A Data Availability Statement is required for all papers reporting original data. This must be in the Additional Information section of the manuscript itself. It must have the paragraph heading 'Data Availability Statement'. All data supporting the results in the paper must be either: in the paper itself; uploaded as Supporting Information for Online Publication; or archived in an appropriate public repository. The statement needs to describe the availability or the absence of shared data. Authors must include in their statement: a link to the repository they have used, or a statement that it is available as Supporting Information; reference the data in the appropriate sections(s) of their manuscript; and cite the data they have shared in the References section. Whenever possible, the scripts and other artefacts used to generate the analyses presented in the paper should also be publicly archived. If sharing data compromises ethical standards or legal requirements then authors are not expected to share it, but must note this in their statement. For more information, see our Statistics Policy.

EDITOR COMMENTS

Reviewing Editor:

Thank you for submitting to The Journal of Physiology. Your manuscript has been reviewed by two expert referees. The authors have addressed the suggested comments. It has been suggested that further reference to the MND data should be added now that the additional data has been included.

Senior Editor:

Thank you for the revised manuscript, which has been considered by the same reviewing editor and referees that considered the original submission. All believe the manuscript has been improved and is potentially acceptable. However, in addition to the new myonuclear domain data in Table 2, the table clearly demonstrates no additional impact of exercise training on any of the reported muscle fibre characteristics reported. The authors should briefly summarise in the Results that no effect of training was detected and also comment on the significance of this to the interpretation of the data in the Discussion. It would also be appropriate to moderate the conclusion in the Abstract that "These novel findings provide supporting evidence for exercise training as a feasible strategy to improve muscle fiber integrity and health in patients with FSHD", which seems to be overstated.

REFEREE COMMENTS

Referee #1:

Thank you to the authors for addressing all of my comments. As the authors have now included MND data (table 2) they may want to briefly summarise that no effect of training was detected in the results section.

Referee #2:

My points have been addressed - well done on the nice manuscript!

END OF COMMENTS

Response to reviewers – Horwath et al. “Increased muscle satellite cell content and preserved telomere length in response to combined exercise training in patients with FSHD”

Reviewing Editor:

Thank you for submitting to The Journal of Physiology. Your manuscript has been reviewed by two expert referees. The authors have addressed the suggested comments. It has been suggested that further reference to the MND data should be added now that the additional data has been included.

Senior Editor:

Thank you for the revised manuscript, which has been considered by the same reviewing editor and referees that considered the original submission. All believe the manuscript has been improved and is potentially acceptable. However, in addition to the new myonuclear domain data in Table 2, the table clearly demonstrates no additional impact of exercise training on any of the reported muscle fibre characteristics reported. The authors should briefly summarise in the Results that no effect of training was detected and also comment on the significance of this to the interpretation of the data in the Discussion. It would also be appropriate to moderate the conclusion in the Abstract that "These novel findings provide supporting evidence for exercise training as a feasible strategy to improve muscle fiber integrity and health in patients with FSHD", which seems to be overstated.

We would like to thank the reviewing editor and senior editor for their guidance throughout the review process. In response to the feedback, we have now included the myonuclear domain data in the Results section and addressed it in the Discussion. Additionally, we have moderated the conclusion made in the abstract.

Referee #1:

Thank you to the authors for addressing all of my comments. As the authors have now included MND data (table 2) they may want to briefly summarise that no effect of training was detected in the results section.

The data on the myonuclear domain has now been added to the result section.

Referee #2:

My points have been addressed - well done on the nice manuscript!

The authors are grateful for the feedback and comments from both reviewers.

Dear Professor Kadi,

Re: JP-RP-2025-287033R2 "Increased muscle satellite cell content and preserved telomere length in response to combined exercise training in patients with FSHD" by Oscar Horwath, Diego Montiel Rojas, Elodie Ponsot, Léonard Féasson, and Fawzi Kadi

We are pleased to tell you that your paper has been accepted for publication in The Journal of Physiology.

Yours sincerely,

Paul Greenhaff
Senior Editor
The Journal of Physiology

If you would like to receive our 'Research Roundup', a monthly newsletter highlighting the cutting-edge research published in The Physiological Society's family of journals (The Journal of Physiology, Experimental Physiology, Physiological Reports, The Journal of Nutritional Physiology and The Journal of Precision Medicine: Health and Disease), please click this link, fill in your name and email address and select 'Research Roundup':
<https://www.physoc.org/journals-and-media/membernews>

- You can help your research get the attention it deserves! Check out Wiley's free Promotion Guide for best-practice recommendations for promoting your work at: www.wileyauthors.com/eoo/guide. You can learn more about Wiley Editing Services which offers professional video, design, and writing services to create shareable video abstracts, infographics, conference posters, lay summaries, and research news stories for your research at: www.wileyauthors.com/eoo/promotion.

Reviewing Editor's comments:

Thank you for submitting to The Journal of Physiology. Your manuscript has been reviewed by two expert referees. The authors have addressed the suggested comments.

Senior Editor's comments:

Thank you for addressing the further comments raised. The manuscript is now deemed to be acceptable for publication. Thank you for selecting The Journal of Physiology to publish your research.

Referee #1:

Thanks for addressing all comments!

Referee #2:

My points have been addressed.

END OF COMMENTS